# Abscisic Acid Improves Rice Thermo-Tolerance by Affecting Trehalose Metabolism

**DOI:** 10.3390/ijms231810615

**Published:** 2022-09-13

**Authors:** Aike Zhu, Juncai Li, Weimeng Fu, Wenting Wang, Longxing Tao, Guanfu Fu, Tingting Chen, Baohua Feng

**Affiliations:** 1National Key Laboratory of Rice Biology, China National Rice Research Institute, Hangzhou 310006, China; 2Nanchong Academy of Agricultural Sciences, Nanchong 637000, China; 3Agronomy College, Jilin Agricultural University, Changchun 130118, China

**Keywords:** rice (*Oryza sativa* L.), heat stress, abscisic acid, trehalose, ATP content

## Abstract

Heat stress that occurs during the flowering stage severely decreases the rice (*Oryza sativa* L.) seed-setting rate. This damage can be reversed by abscisic acid (ABA), through effects on reactive oxygen species, carbohydrate metabolism, and heat shock proteins, but the exact role of trehalose and ATP in this process remains unclear. Two rice genotypes, namely, Zhefu802 (heat-resistant plant, a recurrent parent) and its near-isogenic line (*faded green leaf, Fgl*, heat-sensitive plant), were subjected to 38 °C heat stress after being sprayed with ABA or its biosynthetic inhibitor, fluridone (Flu), at the flowering stage. The results showed that exogenous ABA significantly increased the seed-setting rate of rice under heat stress, by 14.31 and 22.40% in Zhefu802 and *Fgl*, respectively, when compared with the H_2_O treatment. Similarly, exogenous ABA increased trehalose content, key enzyme activities of trehalose metabolism, ATP content, and F1Fo-ATPase activity. Importantly, the opposite results were observed in plants treated with Flu. Therefore, ABA may improve rice thermo-tolerance by affecting trehalose metabolism and ATP consumption.

## 1. Introduction

Rice (*Oryza sativa* L.) is one of the most important food crops worldwide, as it feeds more than half of the world’s population [1]. Rice plants face various kinds of biotic and abiotic stressors in the environment. Heat stress (HS) is occurring more frequently due to global warming. Rice is susceptible to HS, particularly during the reproductive stage. HS occurring during the flowering stage significantly decreases spikelet sterility in rice and can destroy yield [2]. One of the most important ways to improve the rice yield under HS is spraying with a suitable plant growth regulator. Abscisic acid (ABA) is a common regulator used for plants under abiotic stress.

ABA is a “stress hormone” that plays an important role throughout the life cycle of plants, including seed dormancy and germination [3], growth [4], senescence [5], and the responses to abiotic stressors, such as cold [6], drought [7], salinity [8], and HS [9,10]. HS is becoming an increasing threat to rice plants. HS rapidly induces the production of endogenous ABA by plants, which enhances thermo-tolerance by activating the antioxidant system [10,11]. Exogenously applied ABA enhances physiological processes in plants under natural conditions and adverse environmental stress conditions. For example, spraying ABA on plants enhances sucrose synthase (SS) activity and relative starch synthesis gene expression, and improves rice yield [12,13]. Furthermore, exogenous spraying of ABA enhances the thermo-tolerance of rice by increasing carbohydrate contents, including soluble sugars, starch, and non-structural carbohydrate (NSC) contents, in addition to affecting the heat shock protein (HSP) and antioxidant system [14]. Additionally, ABA-deficient mutants are more sensitive to HS than the wild-type [15,16]. Fluridone (Flu) is an inhibitor of phytoene desaturase that inhibits ABA biosynthesis and is reported to reduce plant thermo-tolerance by decreasing the ABA level [17,18]. HSPs and the antioxidant system induced by ABA were initially described to be the main factors improving plant thermo-tolerance [19,20]. However, sugar metabolism and signaling have been recently reported to be important pathways for improving plant thermo-tolerance [21,22]. Sugars are the main source of carbon and energy in plants, where they also play a role as signaling molecules in the regulation of physiological processes [23,24]. A significant decrease in sucrose phosphate synthase (SPS) and SS activities was observed in plants exposed to HS, particularly in heat-sensitive genotypes, which resulted in aborted pollen and lower seed-setting rates [25,26]. Plants have evolved highly complex sugar signaling pathways, including the hexokinase-1, target of rapamycin, and trehalose metabolic pathways [27,28,29].

Trehalose is a non-reducing disaccharide consisting of two glucose residues (a-D-glucopyranosyl-1and 1-a-D-glucopyranoside) that occurs widely. Two enzymatic steps are involved in trehalose synthesis. First, trehalose-6-phosphate synthase (TPS) catalyzes glucose-6-phosphate and UDP-glucose to form trehalose-6-phosphate (T6P). Second, T6P is dephosphorylated by trehalose-6-phosphate phosphatase (TPP) to produce trehalose [30]. Trehalose is involved in many physiological processes, such as carbohydrate storage, transport, and protecting microorganisms and invertebrates from stress [31]. Trehalose content is very low in most plants, suggesting that trehalose may not only participate in metabolism but also participate in signal regulatory pathways [32]. Many studies have shown that trehalose and its precursor T6P are important signaling molecules that regulate plant growth and development, and participate in metabolic regulation and the regulation of gene expression in plants [33,34,35]. Exogenous trehalose inhibits plant growth and changes the carbon allocation [36]. Exogenous trehalose applied with other metabolizable sugars restores the inhibited growth, suggesting that trehalose inhibits plant growth due to impaired sugar utilization rather than excess accumulation of starch [37]. Overexpressing the fused *E. coli* trehalose biosynthetic genes *otsA* and *otsB* in rice increases the soluble sugar level and enhances tolerance to abiotic stress [38,39]. *OsTPP7*/*qAG9-2* may accelerate T6P turnover to increase sink strength in proliferating heterotrophic tissues due to low sugar availability, thus enhancing tolerance to anaerobic germination [40]. However, studies on the role of trehalose in HS are insufficient. Luo et al. [41] observed that exogenous trehalose enhances HS tolerance in winter wheat. An exogenous trehalose pretreatment of plants under HS protects the thylakoid membrane proteins and photosynthetic capacity, reduces electrolyte leakage, malondialdehyde content, superoxide anion, and hydrogen peroxide levels, and inhibits lipoxygenase activity [41]. Furthermore, a proper concentration of exogenous trehalose significantly enhances the photochemical efficiency of photosystem II and increases photochemical quenching in fava beans [42]. *OsTPS8* regulates suberin deposition and soluble sugar accumulation in rice and, thus, confers salinity tolerance by altering ABA-responsive genes [43]. However, the role of trehalose in ABA signaling to induce heat tolerance, has not been investigated.

In the present study, exogenous ABA was applied to Zhefu802 (heat-tolerant) and its pale green near-isogenic line (*faded green leaf, Fgl;* heat-sensitive). Carbohydrate content, carbohydrate key enzymes, trehalose content, key trehalose metabolic enzymes, ATP content, and ATP syntheses were determined to study the role of trehalose in the effects of ABA on the flowering stage of rice under HS.

## 2. Results

### 2.1. Exogenous ABA Improves Rice Thermo-Tolerance 

As shown in Figure 1, no significant differences were observed in the rice seed-setting rates among the three treatments under the control condition. However, applying ABA to plants under HS significantly increased the rice seed-setting rate, whereas applying Flu significantly decreased the seed-setting rate. Decreases in the seed-setting rate of 49.08, 43.74, and 54.44% were observed in Zhefu802 plants treated with H_2_O, ABA, and Flu, respectively, under HS compared with their respective control treatments. However, these changes were 61.20, 56.78, and 74.05% in the *Fgl* plants treated with H_2_O, ABA, and Flu under HS, respectively (Figure 1).

### 2.2. Exogenous ABA Improves Sugar Metabolic Enzyme Activities

As exogenous ABA increased carbohydrate content, we measured the key enzyme activities involved in the sugar metabolic pathway. Spraying exogenous ABA or Flu did not decrease SS activity in either genotype under the control condition. HS significantly decreased SS activity in both genotypes. ABA alleviated this decrease, while Flu aggravated the decrease. The decreases in SS activity were 30, 12, and 40% in Zhefu802 plants treated with H_2_O, ABA, and Flu, respectively, under HS when compared with their respective controls. However, the changes were 20, 18, and 42% in *Fgl* plants treated with H_2_O, ABA, and Flu, respectively, under HS (Figure 2A). These results indicate that exogenous ABA significantly alleviated the decease induced by HS.

Exogenous ABA and Flu did not decrease the soluble acid invertase (S-AI) activity under the control condition in either genotype when compared with the respective H_2_O treatment. HS increased S-AI activity. ABA significantly enhanced this increase, whereas spraying Flu lowered the increase. Increases in S-AI activity of 36.8, 102.1, and 18.4% were observed in Zhefu802 plants treated with H_2_O, ABA, and Flu, respectively, under HS, compared with their respective controls. However, these changes were 37.5, 95.2, and 23.7% in *Fgl* plants treated with H_2_O, ABA, and Flu, respectively, under HS (Figure 2B). These results indicate that exogenous ABA significantly improved the increase induced by HS.

Exogenous ABA and Flu did not decrease soluble starch synthase (SSS) activity under the control condition in either genotype, except Zhefu802 plants treated with ABA had higher SSS activity than the other treatments. HS decreased SSS activity. Plants treated with ABA significantly enhanced the decrease, whereas Flu inhibited the decrease. Decreases in SSS activity of 22.7, 29.4, and 13.7% were observed in Zhefu802 plants treated with H_2_O, ABA, and Flu, respectively, under HS compared with their respective controls. However, these changes were 10.9, 17.3, and 9.2% in *Fgl* plants treated with H_2_O, ABA, and Flu, respectively, under HS (Figure 2C). The results indicated that exogenous ABA significantly improved the decease induced by HS.

### 2.3. Exogenous ABA Improves Trehalose Content

Trehalose is an important component of sugar metabolism, particularly in the regulation of biotic and abiotic stressors, so we measured trehalose content. As shown in Figure 3, exogenous ABA and Flu did not affect the trehalose content under the control condition in either genotype except for Flu-sprayed *Fgl* plants. Spraying Flu on *Fgl* plants significantly decreased trehalose content. However, HS significantly increased trehalose content. Treatment with ABA enhanced this increase, particularly in Zhefu802 plants, whereas exogenous Flu reduced this increase. Increases in trehalose content of 8.5, 55.6, and 8.1% were observed in Zhefu802 plants treated with H_2_O, ABA, and Flu under HS, respectively, compared with their respective controls. However, these changes were 14.8, 17.7, and 5.4% in *Fgl* plants treated with H_2_O, ABA, and Flu, respectively, under HS (Figure 3). These results indicated that exogenous ABA significantly improved the increase induced by HS.

### 2.4. Exogenous ABA Improves the Activity of Key Enzymes Involved in Trehalose Metabolism

As the trehalose content was influenced, we measured the key enzyme involved in trehalose metabolism. As shown in Figure 4, spraying ABA or Flu did not decrease the activities of key trehalose enzymes, including SPS, TPS, and TPP in either genotype under the control condition. HS significantly decreased the activities of these three key enzymes in both genotypes. ABA alleviated the decrease, whereas Flu aggravated the decrease. The decreases in the activity of SPS were 38.5, 29.3, and 42.0% in Zhefu802 plants treated with H_2_O, ABA, and Flu, respectively, under HS compared with their respective controls. However, these changes were 50.1, 30.8, and 57.7% in *Fgl* plants treated with H_2_O, ABA, and Flu, respectively, under HS (Figure 4A). Decreases in TPS activity of 36.4, 15.1, and 57.9% were observed in Zhefu802 plants treated with H_2_O, ABA, and Flu, respectively, under HS when compared with their respective controls. However, these changes were 50.1, 30.8, and 57.7% in *Fgl* plants treated with H_2_O, ABA, and Flu, respectively, under HS (Figure 4B). Decreases in TPP activity of 43.1, 32.3, and 56.3% were observed in Zhefu802 plants treated with H_2_O, ABA, and Flu under HS, respectively, compared with their respective controls. However, these changes were 59.3, 52.3, and 66.0% in *Fgl* plants treated with H_2_O, ABA, and Flu, respectively, under HS (Figure 4C). The results indicated that exogenous ABA significantly alleviated the decrease induced by HS.

### 2.5. Exogenous ABA Improves ATP Content

Energy plays a key role in regulating abiotic stress and plant growth. As shown in Figure 5, no significant difference in ATP content was observed in Zhefu802 plants among the three treatments under the control condition. Exogenously spaying ABA decreased ATP content in *Fgl* plants, whereas Flu increased ATP content. HS significantly increased ATP content in both genotypes. Increases in the ATP levels of 68.7, 36.1, and 93.5% were observed in Zhefu802 plants treated with H_2_O, ABA, and Flu, respectively, when compared with their respective controls. However, these changes were 52.1, 48.4, and 62.1%, respectively, in *Fgl* plants (Figure 5). These results demonstrated that exogenous ABA significantly alleviated the increase induced by HS, suggesting that ATP utilization was much more fluent than the H_2_O treatment.

### 2.6. Exogenous ABA Improves F1Fo-ATPase Activity

As F1Fo-ATPase is an important enzyme that catalyzes the terminal step in oxidative phosphorylation, which determines the ATP production, we measured the F1Fo-ATPase activity. No significant difference in F1Fo-ATPase activity was observed among the three treatments in either genotype under the control condition. HS significantly decreased F1Fo-ATPase activity in both genotypes. Decreases in the F1Fo-ATPase activity levels of 47.9, 13.2, and 52.1% were observed in Zhefu802 plants treated with H_2_O, ABA, and Flu, respectively, when compared with their respective controls. However, these changes were 35.5, 22.8, and 49.6%, respectively, in *Fgl* plants (Figure 6).

## 3. Discussion

Our results indicate that ABA enhanced the rice seed-setting rate caused by HS (Figure 1), which was consistent with previous research that ABA confers heat resistance to plants [10,44]. It also has been documented that ABA signaling deficient mutant (*abi1-1*) and the ABA biosynthetic mutant (*aba1-1*) are extremely sensitive to HS [44]. Additionally, exogenous ABA enhances the tolerance of rice to HS during the entire growth period [45], indicating that exogenous ABA enhances rice thermo-tolerance.

Our previous study indicated that exogenous ABA alleviated heat damage and significantly increased rice spikelet fertility. Carbohydrate contents, including those of NSC, soluble sugars (sucrose, glucose, and fructose), and starch, were significantly higher in spikelets in the ABA treatment than those in the H_2_O and Flu treatments in both genotypes [14]. Islam et al. [14] deduced that sucrose transport and metabolism-related genes, including *sucrose transporter* (*SUT*), *invertase* (*INV*), and *sucrose synthase* (*SUS*) genes, are induced by ABA under HS. This is consistent with our present study, which showed damage to carbohydrate-related enzymes, including SS, S-AI, and SSS activity induced by HS, was alleviated by the exogenous ABA application and was accelerated by exogenous Flu (Figure 2). It has been well documented that ABA enhances sucrose unloading and metabolism in rice spikelets under HS and drought stress [46,47,48]. The mechanism of sucrose transport and metabolism induced by ABA under HS may be through increasing *SUT*, *INV*, and *SUS* gene expression levels [21,49]. HS significantly decreases Rubisco (carbon-fixing enzyme) and SS activities in rice leaves [50]. In addition, the sucrose-cleaving enzyme INV is also inhibited along with SPS and SS under HS [51]. We concluded that exogenous ABA alleviated the damage to SS, S-AI, and SSS activity induced by HS. The trehalose pathway is one of the most important components of sugar metabolism, as it plays an important role in regulating plant growth and response to abiotic stresses.

Trehalose is degraded into two glucose residues by trehalase [52]. It has been reported that trehalose accumulation enhances the tolerance of crop plants to abiotic stressors [35,40]. One study reported that it was the precursor T6P, not trehalose, that plays a key role in regulating sugar allocation between the source and sink in plants [35]. Trehalose and T6P have been thought to be involved in carbon availability, and the T6P levels may be more important and rapidly affect the response to exogenous sucrose [53]. Griffiths et al. [54] reported that promoting the T6P pathway by chemical intervention enhances plant performance and function. Li et al. [55] deduced that the accumulation of T6P increases sugar consumption by enhancing sugar allocation to sink organs, thereby decreasing sugar levels in source tissues and maintaining sugar homeostasis. We did not measure the T6P content in the present study. Additionally, we did not conduct a T6P application experiment. HS increased trehalose content, particularly in *Fgl*, whereas treatment with exogenous ABA enhanced this increase, and Flu inhibited the increase (Figure 3). TPP and TPS are two major regulators of trehalose metabolism. Regulating the activities of these two enzymes can affect plant growth and abiotic stress responses. Overexpressing *OsTPS8* enhances salinity tolerance without any yield penalty, suggesting its significance in the genetic improvement of rice [43]. *OsTPS1* overexpressing lines enhance the response of rice to various abiotic stressors during the seedling stage by increasing trehalose and related gene levels [56,57,58,59]. Overexpressing *OsTPP1* enhances the tolerance of rice to low temperatures by increasing the trehalose level [60]. ABA significantly increases *TPS1* and *TPP7* levels in rice grains [48], which promotes trehalose synthesis and enhances tolerance to abiotic stressors [40,61]. In our study, exogenous application of ABA significantly alleviated the decrease in the SPS, TPS, and TPP activities induced by HS, whereas applying Flu aggravated the decrease (Figure 4). Plant growth and development consumes a tremendous amount of ATP, particularly in plants under abiotic stress [62]. As a fade-back mechanism, more ATP is consumed for resistance to abiotic stress such as cleaving ROS, enhancing antioxidant capacity, and accumulating heat shock proteins [14]. In the present study, higher ATP content was observed in the HS than the control treatment. However, lower ATP content was observed in ABA-treated plants, whereas higher ATP content was observed in Flu-treated plants (Figure 5). These results suggest that the ATP consumption was changed. Mitochondrial respiration provides ATP for plant growth and development, and five complexes in the alternative oxidase system and the photosystems are involved in respiration [63,64]. During these five complexes, complex V (F1Fo-ATP synthase) is the most important complex that catalyzes the terminal step in oxidative phosphorylation, converting the electrochemical gradient across the inner membrane into ATP for cellular biosynthesis, which determines the ATP production [63]. These proteins are frequently affected by abiotic stressors, such as HS [65,66]. It also has been documented that, under high temperatures, the energy produced by respiration is very important for crop growth and development [62]. In the present study, F1Fo-ATPase decreased in response to HS, whereas applying ABA significantly alleviated the decrease, and the Flu treatment aggravated the decrease (Figure 6). This result illustrates that exogenous ABA enhances the ATP consumption, while Flu attenuates it, which is inconsistent with previous studies. Chen et al. [48] reported that ABA alleviates the inhibitory effect of HS, as higher ATP content was observed in ABA-treated plants than in those treated with H_2_O or Flu. These findings suggest that exogenous application of ABA enhances trehalose metabolism and ATP consumption.

## 4. Materials and Methods

### 4.1. Plant Materials and Growing Conditions

Two rice genotypes Zhefu802 (the recurrent parent, heat sensitive) and its near-isogenic line (*faded green leaf, Fgl;* heat resistant) were used in this study. The experiment was conducted at the Chinese National Rice Research Institute (Hangzhou, Zhejiang Province, China) during 2018–2019. Round black buckets (10 cm diameter and 20 cm height) were filled with 10 kg of dry paddy soil containing 1400 mg/kg total N, 1470 mg/kg total P, 400 mg/kg total K, 33.74 mg/kg available N, 10.8 mg/kg available P, 58.4 mg/kg available K, and 0.99% organic matter. The seeds were soaked in the dark for 48 h and then sprouted at 37 °C for 24 h. Ten seeds were directly sown per pot and thinned to three plants per pot with 36 repetitions at the 4-leaf stage. The plants were cultivated under natural conditions until the pollen mother cell meiotic stage. Then, all rice plants were divided into three groups; two groups were sprayed with 100 µM ABA and 100 μM Flu once, respectively, and treated with double-distilled water as a control treatment. The rice plants were separated into two groups 2 h after spraying. One group was placed in a growth chamber with HS (39–41 °C from 09:00 to 15:00 and 30 °C at night), while the other group was subjected to the control chamber (30–32 °C during the day and 24 °C at night). After 7 days of HS treatment, all of the rice plants were placed outside under natural conditions until they matured. Before they were removed from the chambers, all of the panicles in the three pots of each treatment were sampled for physiological and biochemical analyses, including the key enzyme activities involved in sugar and trehalose metabolism, ATP content, trehalose content, and ATPase activity.

### 4.2. Spikelet Fertility

After the plants had matured completely, all of the panicles in the remaining three pots were sampled. To calculate the setting rate, all unfilled grains and filled grains were numbered after drying in an oven at 50 °C for 48 h.

### 4.3. Measurement of the Key Enzymes Involved in Sugar Metabolism

SS, S-AI, and SSS were extracted following the manufacturer’s instructions (Cominbio Co. Ltd., (Suzhou, China) [51]. About 0.1 g of spikelets was ground into a powder and homogenized with 1 mL of an extraction buffer containing 50 mM HEPES (pH 7.5), 5 mM MgCl_2_, 1 mM EDTA-Na_2_, 0.5 mM dithiothreitol, 1% Triton X−100, 2% polyvinyl pyrrolidone, and 10% glycerol. The suspension was immediately centrifuged at 8000× *g* for 10 min at 4 °C. The supernatant was collected to measure enzyme activities. After some reaction steps, the absorbance values at 510 nm for SS, 540 nm for S-AI, and 480 nm for SSS were recorded to calculate the enzyme activities.

### 4.4. Measurement of Trehalose Content

Trehalose content was determined using a test kit from Cominbio Co. Ltd. About 0.1 g of spikelets was ground into a powder and homogenized with 1 mL of extraction buffer. The suspension was held at room temperature for 45 min, shaken 3–5 times, and centrifuged at 8000× *g* for 10 min at 25 °C. The supernatant was collected to measure trehalose content. After the reaction step, absorbance at 620 nm was determined to calculate the trehalose content.

### 4.5. Measurement of Key Enzyme Activities Involved in Trehalose Metabolism

SPS, TPS, and TPP were extracted using test kits from Cominbio Co. Ltd. About 0.1 g of spikelets was ground into a powder and homogenized with 1 mL of the extraction buffer provided with the kit. The suspension was immediately centrifuged at 8000× *g* for 10 min at 4 °C. The supernatant was collected to measure enzyme activity. After the reaction steps, the absorbance values at 480 nm for SPS, 340 nm for TPS, and 660 nm for TPP were recorded to calculate the enzyme activities.

### 4.6. Measurement of ATP Content

ATP content was measured using a test kit from Cominbio Co. Ltd. [51]. A 0.05 g portion of spikelets was ground into a powder, homogenized with 0.5 mL of acid extraction buffer in an ice bath, and centrifuged at 8000× *g* for 10 min at 4 °C. The supernatant was collected and mixed with an isopycnic alkaline extraction buffer. Then, the miscible liquid was centrifuged at 8000× *g* for 10 min at 4 °C. The supernatant was collected to measure ATP content. After some reaction steps, the absorbance value at 700 nm was used for the calculation.

### 4.7. Measurement of F1Fo-ATPase Activity

F1Fo-ATPase activity was measured using a test kit from Cominbio Co. Ltd. [67]. About 0.1 g of spikelets was ground into a powder, homogenized with 1 mL of extraction buffer (50 mM Tris-HCl, 5 mM MgCl_2_, 5 mM K_2_HPO_4_, 2 mM ADP, and 10% glycerol, pH 8.0), and incubated at 37 °C for 5 min. The supernatant was collected to measure FoF1-ATPase activity. The ATP synthesis reaction was stopped by adding a one-tenth volume of 4% TCA, and then 3 μL of the liquor was diluted with 100 μL of the assay mix dilution buffer. Absorbance at 660 nm was used to calculate F0F1-ATPase activity.

### 4.8. Statistical Analyses

The data were analyzed using SPSS 11.5 (SPSS Inc., Chicago, IL, USA) and Excel 2010 software (Microsoft Inc., Redmond, WA, USA). Three independent experiments were conducted. The mean values and standard deviations in the figures represent data from three replicates. Two-way analysis of variance (ANOVA) with two factors (temperature and treatment) were used to compare the differences in LSD test with *p* (*p* ≤ 0.05).

## 5. Conclusions

Based on our previous study, exogenous application of ABA enhances sucrose transport and accelerates sucrose metabolism to maintain the carbon balance and energy homeostasis, thus improving rice thermos-tolerance. Here we confirmed that exogenous application of ABA enhances the rice seed-setting rate under HS. We also measured the key enzymes involved in sugar metabolism to verify the role of carbohydrate metabolism under HS. Furthermore, we found the trehalose content and key enzymes involved in trehalose metabolism were influenced, which may indicate the important role of trehalose in the ABA-enhancing thermo-tolerance process. Additionally, ATP content decreased whereas F1Fo-ATPase activity increased in the ABA treatment under HS, which may indicate that the ATP consumption was sharply increased. Hence, we conclude that exogenous ABA may significantly increase the rice seed-setting rate under HS through trehalose metabolism and ATP consumption.

## Figures and Tables

**Figure 1 ijms-23-10615-f001:**
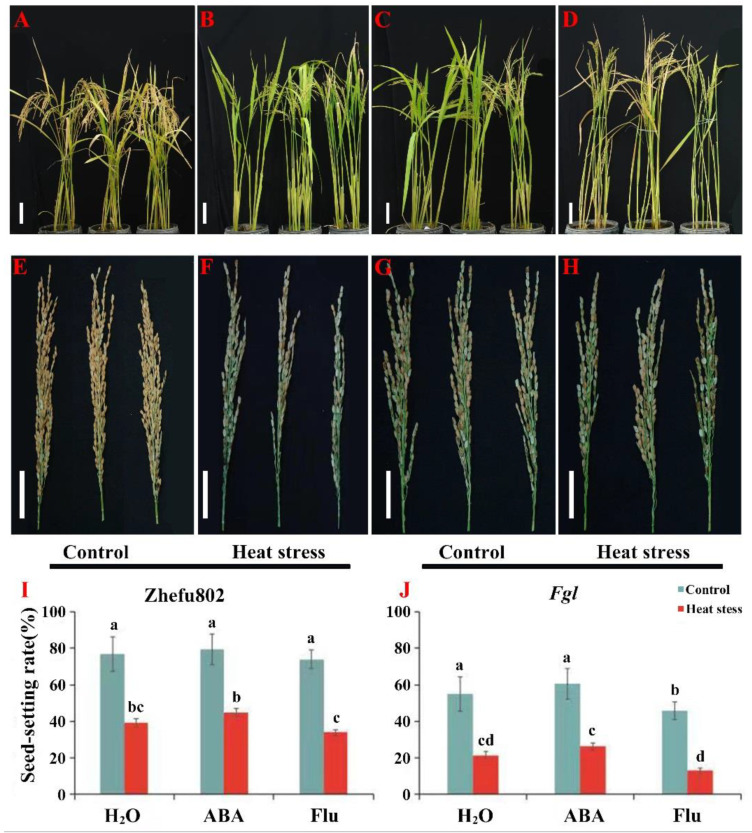
The effects of exogenous ABA on the seed-setting rate of rice under heat stress. (**A**,**B**,**E**,**F**) are Zhefu802 plants. (**C**,**D**,**G**,**H**) are *Fgl* plants. (**I**,**J**) are the rice seed-setting rate for Zhefu802 and *Fgl* respectively. ABA, abscisic acid. Flu, the ABA synthase inhibitor fluridone. In (**A**–**H**), three pots or three panicles were selected from H_2_O, ABA, and Flu treatment in sequence. The reference scale from (**A**–**D**) and (**E**–**H**) represents 10 and 5 cm respectively. Error bars denote standard deviations (*n* = 3). Different letters indicate significant differences between the control and heat stress treatments within a genotype by two-way analysis of variance (temperature and treatment) (*p* < 0.05).

**Figure 2 ijms-23-10615-f002:**
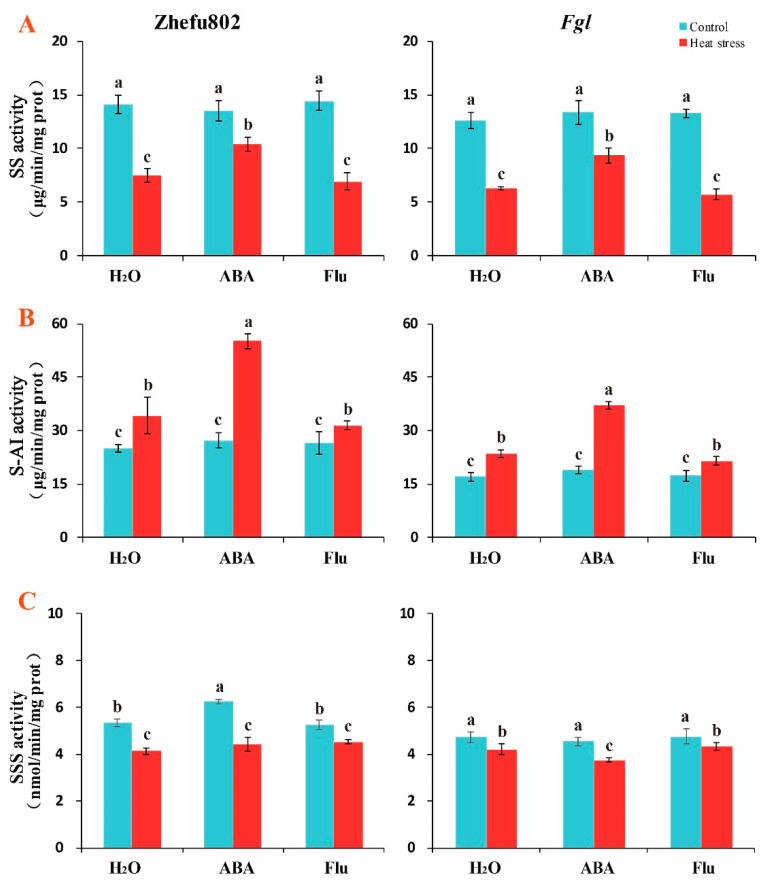
The effects of exogenous ABA on the key enzymes of sugar metabolism in rice plants under heat stress. (**A**), SS activity; SS, sucrose synthase. (**B**), S-AI activity; S-AI, acid invertase. (**C**), SSS activity; SSS, soluble starch synthase. ABA, abscisic acid. Flu, the ABA synthase inhibitor fluridone. Error bars denote standard deviations (*n* = 3). Different letters indicate significant differences between the control and heat stress treatments within a genotype by two-way analysis of variance (temperature and treatment) (*p* < 0.05).

**Figure 3 ijms-23-10615-f003:**
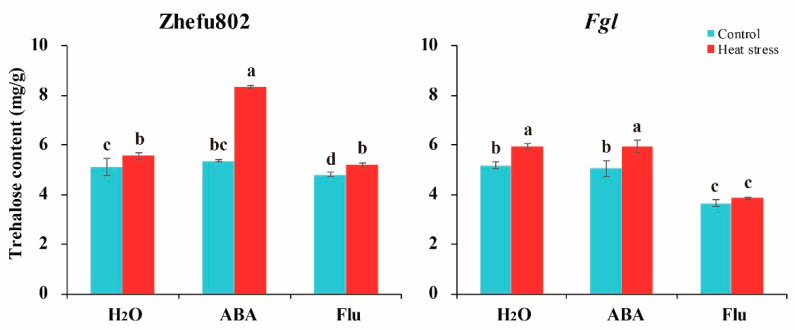
The effects of exogenously applied ABA on trehalose content of rice plants under heat stress. ABA, abscisic acid. Flu, the ABA synthase inhibitor fluridone. Error bars denote standard deviations (*n* = 3). Different letters indicate significant differences between the control and heat stress treatments within a genotype by two-way analysis of variance (temperature and treatment) (*p* < 0.05).

**Figure 4 ijms-23-10615-f004:**
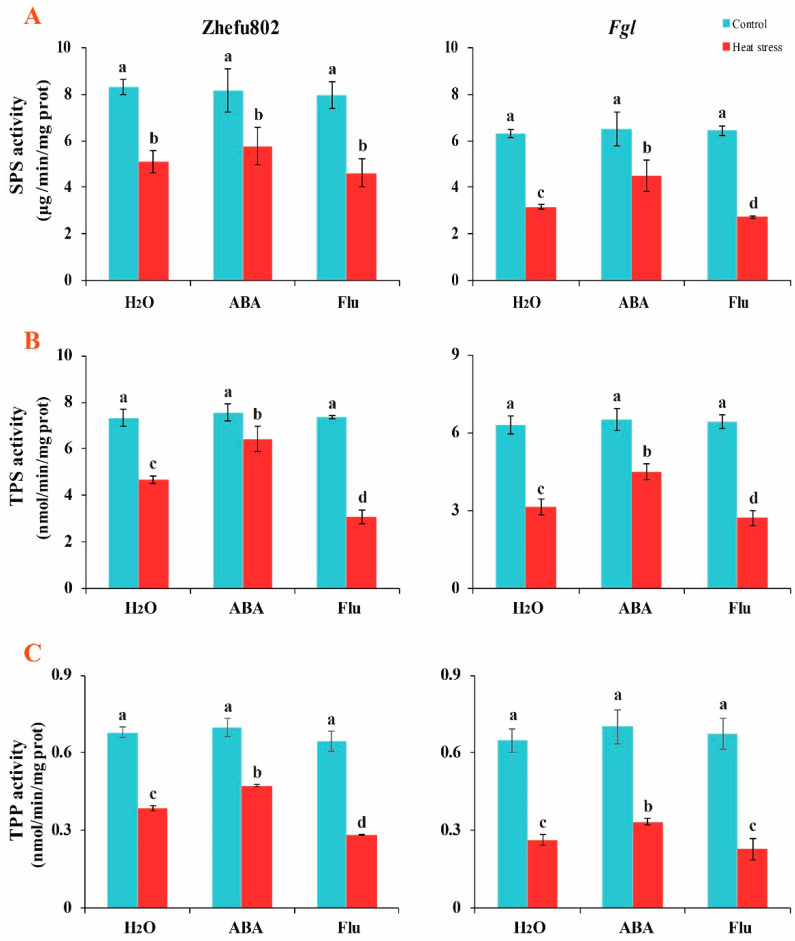
The effects of exogenous ABA on the key enzymes of trehalose metabolism in rice plants under heat stress. (**A**), SPS activity; SPS, sucrose phosphate synthase. (**B**), TPS activity; TPS, trehalose-6-phosphate synthase. (**C**), TPP activity; TPP, trehalose-6-phosphate phosphatase. ABA, abscisic acid. Flu, the ABA synthase inhibitor fluridone. Error bars denote standard deviations (*n* = 3). Different letters indicate significant differences between the control and heat stress treatments within a genotype by two-way analysis of variance (temperature and treatment) (*p* < 0.05).

**Figure 5 ijms-23-10615-f005:**
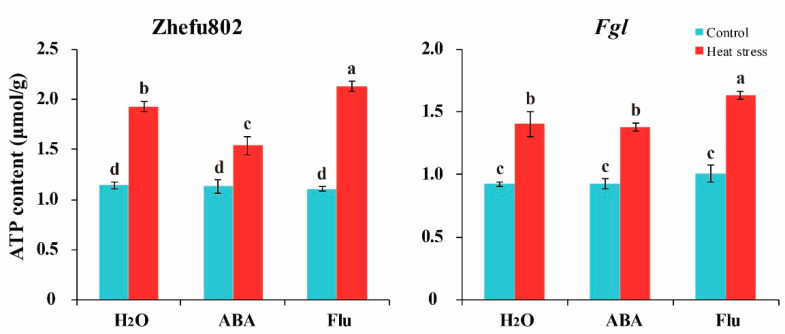
The effects of exogenously applied ABA on ATP content of rice plants under heat stress. ABA, abscisic acid. Flu, the ABA synthase inhibitor fluridone. Error bars denote standard deviations (*n* = 3). Different letters indicate significant differences between the control and heat stress treatments within a genotype by two-way analysis of variance (temperature and treatment) (*p* < 0.05).

**Figure 6 ijms-23-10615-f006:**
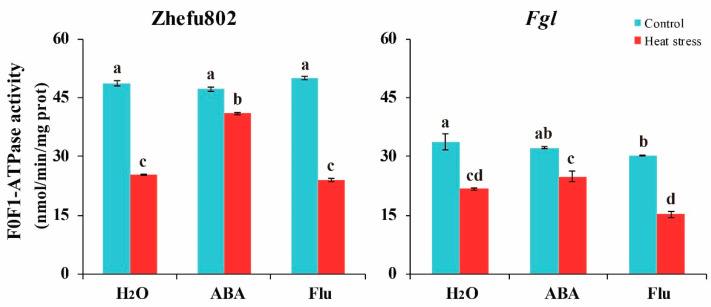
The effects of exogenously applied ABA on F1Fo-ATPase activity of rice plants under heat stress. ABA, abscisic acid. Flu, the ABA synthase inhibitor fluridone. Error bars denote standard deviations (*n* = 3). Different letters indicate significant differences between the control and heat stress treatments within a genotype by two-way analysis of variance (temperature and treatment) (*p* < 0.05).

## Data Availability

Not applicable.

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
