# Peer review of "Abscisic Acid Improves Rice Thermo-Tolerance by Affecting Trehalose Metabolism"

_ijms, 2022, doi:10.3390/ijms231810615_

Round 1

Reviewer 1 Report

The authors claimed that ABA-induced trehalose synthesis enhanced energy use efficiency, and lead to heat stress tolerance in rice. A trial was conducted to study heat-stressed rice responses to ABA treatment and ABA-inhibitor treatment- Fluridone. The experimental design was focused on the role of ABA in heat stress resistance by assessing trehalose synthesis, ATP synthesis, and carbohydrate synthesis. However, the experimental design did not cover the evaluation of the relationship between trehalose and energy use efficiency. The results do not support the research question stated in the manuscript- evaluation of the role of trehalose in heat stress tolerance. Authors may consider redefining the research question and revise the manuscript accordingly.

-         Energy use efficiency refers to the usage of an energy source to produce a certain amount of product (such as WUE in photosynthesis). The results did not cover the source-sink relation.

Result

-          Figure 1. A-H Are these plants selected from specific treatment (H2O, ABA, Flu)? Or 1 from each treatment? This should be stated in figure or legend. The reference scale is missing. Pictures’ alignment needs improvement.

-          Figure 2 Subfigures are poorly aligned. The font size is not uniform. Why are these parameters measured? A brief introduction would be good before Line 110. What are the numbers and % present in the result mean? Are they matching your objective? A brief to explain the result would be good.

-          Figure 3. H2O vs Flu treatment result showed that reduction in Trehalose content after HS was higher in sensitive cultivar (Fgl). The resistant cultivar (Zhefu802) showed barely any change in Trehalose content. ABA treatment only increased Trehalose content in Zhefu802. Is this mean that ABA and trehalose were not important in HS resistance based on the response of sensitive and tolerant cultivars to ABA and Flu treatments?

Axis units were bolded but the y-axis label was not bolded. Looks weird.

-          Figure 4 Please check the alignment and font size. Same issue as in Figure 2. No intro of the measurement intention. No explanation of the findings.

-          Figure 5 Font size- Fgl. Intro and explanation missing

Statistic

-          Line311What do the 3 independent experiments means here? This should be stated in the plant material part. If each treatment (H2O, ABA, and FLU) was measured in 3 separate trials, the results were not comparable. Even if the trials were conducted in a controlled environment, daylight and temperature variation in different seasons from 2018-2019 will still affect the growth of rice. The one-way ANOVA of each treatment doesn’t mean anything as rice response to HS is not the objective. Two-way ANOVA (HS, ABA treatments) with posthoc analysis should be conducted.

Author Response

Comments and Suggestions for Authors

Q1: The authors claimed that ABA-induced trehalose synthesis enhanced energy use efficiency, and lead to heat stress tolerance in rice. A trial was conducted to study heat-stressed rice responses to ABA treatment and ABA-inhibitor treatment- Fluridone. The experimental design was focused on the role of ABA in heat stress resistance by assessing trehalose synthesis, ATP synthesis, and carbohydrate synthesis. However, the experimental design did not cover the evaluation of the relationship between trehalose and energy use efficiency. The results do not support the research question stated in the manuscript- evaluation of the role of trehalose in heat stress tolerance. Authors may consider redefining the research question and revise the manuscript accordingly.

A: Thank you very much for your valuable suggestion, and we have tried our best to revise this manuscript. We used ABA application to improve rice thermotolerance, and during this process, we found trehalose pathway was influenced, also the ATP utilization was increased. So we deduced that ABA improves rice thermotolerance by affecting ATP utilization through trehalose metabolism under heat stress. Maybe this viewpoint is correct. Thank you again.

Q2: Energy use efficiency refers to the usage of an energy source to produce a certain amount of product (such as WUE in photosynthesis). The results did not cover the source-sink relation.

A: Thank you very much for your valuable suggestion. I totally agreed that the definition of the energy use efficiency. The reviewer was right about the source data absence. Actually, this MS was a further research based on our present study (the reference 14), treatment was the same. So I will cite this data as a supplementary to confirm the source problem. Thank you again.

Q3: Figure 1. A-H Are these plants selected from specific treatment (H2O, ABA, Flu)? Or 1 from each treatment? This should be stated in figure or legend. The reference scale is missing. Pictures’ alignment needs improvement.

A: Thank you very much for your valuable suggestion. I am so sorry to make the reviewer unclear. Actually, A-H from Fig.1 were collected during the same experiment, three pots or three panicles were selected from H2O, ABA and Flu in sequence. Also I have added this information in the figure legend in red to make it clear, and the figure alignment was corrected, also the reference scale has been added. Thank you again.

Q4: Figure 2 Subfigures are poorly aligned. The font size is not uniform. Why are these parameters measured? A brief introduction would be good before Line 110. What are the numbers and % present in the result mean? Are they matching your objective? A brief to explain the result would be good.

A: Thank you very much for your valuable suggestion. I am so sorry to make this mistakes. In the revised MS, I have already aligned the figures, also uniformed the font size. And also, I have added a brief introduction in red to explain the reason why we measure this parameters. An explanation for the results also has been added in red. Thank you again.

Q5: Figure 3. H2O vs Flu treatment result showed that reduction in Trehalose content after HS was higher in sensitive cultivar (Fgl). The resistant cultivar (Zhefu802) showed barely any change in Trehalose content. ABA treatment only increased Trehalose content in Zhefu802. Is this mean that ABA and trehalose were not important in HS resistance based on the response of sensitive and tolerant cultivars to ABA and Flu treatments? Axis units were bolded but the y-axis label was not bolded. Looks weird.

A: Thank you very much for your excellent query. You doubt that the reduction in trehalose content after HS is higher in Fgl than Zhefu802, which might mean the nonsense of trehalose in HS. But this induced carefully thinking, as the ABA spraying could significant increase the rice seed-seting rate, which indicated ABA play an important role in improving rice thermotolerance. But the trehalose content in response of Zhefu802 and Fgl? As the increasing range is 8.5% and 14.8% in Zhefu802 and Fgl respectively. I preferred that the dose reaction but not the irrelevant was the key point. And during the further study, I will try my best to verify this hypothesis. By the way, the Axis bolded problem was corrected in the revised MS.

Q6: Figure 4 Please check the alignment and font size. Same issue as in Figure 2. No intro of the measurement intention. No explanation of the findings.

A: Thank you very much for your valuable suggestion. Sorry for this mistakes. In the revised MS, I have already aligned the figures and uniformed the font size. Also, a brief introduction and an explanation has been added in red. Thank you again.

Q7: Figure 5 Font size- Fgl. Intro and explanation missing

A: Thank you very much for your valuable suggestion. Sorry for this mistakes. In the revised MS, I have already aligned the figures and uniformed the font size. Also, a brief introduction and an explanation has been added in red. Thank you again.

Q8: Line311 What do the 3 independent experiments means here? This should be stated in the plant material part. If each treatment (H2O, ABA, and FLU) was measured in 3 separate trials, the results were not comparable. Even if the trials were conducted in a controlled environment, daylight and temperature variation in different seasons from 2018-2019 will still affect the growth of rice. The one-way ANOVA of each treatment doesn’t mean anything as rice response to HS is not the objective. Two-way ANOVA (HS, ABA treatments) with posthoc analysis should be conducted.

A: Thank you very much for your valuable suggestion. Sorry for this obscure statement. Actually, three independent experiments means we did this experiment three times during 2018-2019, also we conducted all the three treatments under HS and control condition each time. We obtained the same tendency result, so, we draw the conclusion. By the way, in the revised MS, two-way ANOVA analysis was used to substitute the original resulst. Thank you again.

Reviewer 2 Report

Although this manuscript is interesting and well explained but English language should be revised throughout the manuscript. Manuscript contains some typing errors and the literature contains a few typing mistakes. Overall, I believe this work to be well planned, fitting the scope of the journal, but substantial revision is necessary to make this paper more presentable.

·     The abstract must include numerical results. Conclusive statement with the application of work should be established at the end.

      Add rice scientific name.

    The introduction feels jumpy as there should be narrative links between sentences and paragraphs so that the reader is able to follow your argument. 

      Please add detailed growth conditions during the experiment.

      How old the seedlings were at the time of transplantation?

      It was not clear the age of the plants at the end of the experiment.

      How many time the foliar spray treatments were used?

·     Did the authors applied any fertilizer during experiment? What was the nutrient source?

Authors must provide physio-chemical properties of soil used in current study.

It is suggested to add more details regarding planting and treatment.

How many biological and technical replicates have you used?

·   The result just presented in a simple way, the relationship between parameters and the treatment can make the results more interesting.

     Discussion is not merely a collection of references to be presented in words only. It must reflect mechanistic depth to support findings. I therefore, advice authors to revise full discussion segment and draw conclusive and parallel lines to make it easy and readily understandable for readers.

     The conclusion section should be rewritten to depict the correct findings as well as some future prospects.

     Please write the complete term before using its abbreviation. Afterwards write abbreviation of that term in question. However, do not write an abbreviation at the start of a sentence.

     Double check that all references are cited within the text, and that all citations within the text have a corresponding reference.

     Please go through the whole reference list and incorporate the required changes according to journals’ author guidelines.

Author Response

Comments and Suggestions for Authors

Q1: Although this manuscript is interesting and well explained but English language should be revised throughout the manuscript. Manuscript contains some typing errors and the literature contains a few typing mistakes. Overall, I believe this work to be well planned, fitting the scope of the journal, but substantial revision is necessary to make this paper more presentable.

A: Thank you very much for your valuable suggestion, and we are sorry for the careless errors. In the revised manuscript, we have checked the manuscript carefully with the help of the language editing company (Editorbar, https://www.editorbar.com/index), which has been highlighted in red. Thank you again.

Q2: The abstract must include numerical results. Conclusive statement with the application of work should be established at the end.

A: Thank you very much for your valuable suggestion. In the revised MS, we have added some numerical results in the abstract and the sense of this work which has been highlighted in red.

Q3: Add rice scientific name.

A: Thank you very much for your valuable suggestion. In the revised MS, we have added rice scientific name which has been highlighted in red.

Q4: The introduction feels jumpy as there should be narrative links between sentences and paragraphs so that the reader is able to follow your argument.

A: Thank you very much for your valuable suggestion, and we are sorry for this obscure expression. In the revised MS, we have added transition sentences to make the MS readably.

Q5: Please add detailed growth conditions during the experiment. How old the seedlings were at the time of transplantation? It was not clear the age of the plants at the end of the experiment. How many time the foliar spray treatments were used?

A: Thank you very much for your valuable suggestion, and we are sorry for this obscure expression. In the revised MS, we have added the plant growth condition in much more details which have been highlighted in red. Actually, we did not transplant the rice plants, we used direct seeding rice instead and thinned to three plants per pot at 4-leaf stage which has been highlighted in red. We conducted this explication only once during the experiment. Sampling was conducted before moving out of the chamber, and other pots were till matured.

Q6: Did the authors applied any fertilizer during experiment? What was the nutrient source? Authors must provide physio-chemical properties of soil used in current study. It is suggested to add more details regarding planting and treatment. How many biological and technical replicates have you used?

A: Thank you very much for your valuable suggestion, and we are sorry for this obscure expression. Actually, we fertilized these plants at a normal regulation, we did this whole experiments three times during 2018-2019, and also at each time, three pots were conducted. This means we used three biological replicates and three technical replicates. In the revised MS, we have added the plant growth condition in much more details which have been highlighted in red.

Q7: The result just presented in a simple way, the relationship between parameters and the treatment can make the results more interesting.

A: Thank you very much for your valuable suggestion. In the revised MS, we add some sentences to explain the relationship between parameters and treatment.

Q8: Discussion is not merely a collection of references to be presented in words only. It must reflect mechanistic depth to support findings. I therefore, advice authors to revise full discussion segment and draw conclusive and parallel lines to make it easy and readily understandable for readers.

A: Thank you very much for your valuable suggestion. In the revised MS, we rewritten the whole discussion which has been highlighted in red. But it is a pity that we did not add a mechanism picture because the MS based on some previous research. If we add the mechanism picture, it will be some repetition with the previous study.

Q9: The conclusion section should be rewritten to depict the correct findings as well as some future prospects.

A: Thank you very much for your valuable suggestion. In the revised MS, we rewritten the whole conclusion which has been highlighted in red.

Q10: Please write the complete term before using its abbreviation. Afterwards write abbreviation of that term in question. However, do not write an abbreviation at the start of a sentence.

A: Thank you very much for your valuable suggestion, sorry for these careless errors. In the revised MS, we have corrected all this abbreviation problems.

Q11: Double check that all references are cited within the text, and that all citations within the text have a corresponding reference. Please go through the whole reference list and incorporate the required changes according to journals’ author guidelines.

A: Thank you very much for your valuable suggestion, and we are sorry for these careless errors. In the revised MS, we have already checked the reference list carefully. Thank you again.

Round 2

Reviewer 1 Report

Overall, the revised manuscript is good. Authors made significant edits to the Figures, statistics and writing. However, the authors highlighted the role of trehalose in HS tolerance via controlling ATP utilization in the title/abstract/conclusion, this statement is poorly supported.

Firstly, the positive effect of exogenous ABA treatment on HS resistance was not obvious based on the seed setting rate result. This does not support the statement ABA improves thermo-tolerance.

Secondly, ABA is a hormone that affects many physiological activities. The improvement of ATP utilization could be affected by other factors regulated by ABA.  If the treatment was exogenous trehalose treatment or ABA treatment with trehalose-inhibitor (rather than ABA inhibitor) then this statement could be proved.

In Figure 3, the trehalose content only slightly increased in H2O condition. The FLU treatment further supported this finding (low ABA in HS-affected rice).  Exogenous ABA only increased trehalose content in the resistance plant. These two findings are interesting and further experiments may be helpful to reveal the role of trehalose in rice under HS.

The findings did support the positive effect of exogenous ABA treatment to

- trehalose content in HS resistance cultivar 

- trehalose enzyme activities

- ATP content and F0F1-ATPase (how are they related to ATP utilization is not clearly interpreted and explained)

I suggest authors revise the title/abstract/conclusion based on the findings.

Minorly,

2.6

What is the role of F0F1-ATPase?? Any relation between F0F1-ATPase and ATP content? Lack of interpretation….. 

2nd para of discussion

SUS, SUT – are they short forms of genes? What is the full name? italic form?

3rd para 

Overexpressing TPS >italic?

Author Response

Comments and Suggestions for Authors

Q1: Overall, the revised manuscript is good. Authors made significant edits to the Figures, statistics and writing. However, the authors highlighted the role of trehalose in HS tolerance via controlling ATP utilization in the title/abstract/conclusion, this statement is poorly supported.

Firstly, the positive effect of exogenous ABA treatment on HS resistance was not obvious based on the seed setting rate result. This does not support the statement ABA improves thermo-tolerance.

Secondly, ABA is a hormone that affects many physiological activities. The improvement of ATP utilization could be affected by other factors regulated by ABA. If the treatment was exogenous trehalose treatment or ABA treatment with trehalose-inhibitor (rather than ABA inhibitor) then this statement could be proved.

In Figure 3, the trehalose content only slightly increased in H2O condition. The FLU treatment further supported this finding (low ABA in HS-affected rice). Exogenous ABA only increased trehalose content in the resistance plant. These two findings are interesting and further experiments may be helpful to reveal the role of trehalose in rice under HS.

The findings did support the positive effect of exogenous ABA treatment to

- trehalose content in HS resistance cultivar

- trehalose enzyme activities

- ATP content and F0F1-ATPase (how are they related to ATP utilization is not clearly interpreted and explained)

I suggest authors revise the title/abstract/conclusion based on the findings.

A: Thank you very much for your positive evaluation and valuable suggestion. I totally agree with the reviewers about the absence of trehalose treatment data. It is a pity that we cannot add this experiment this time, but further we will design the experiment much scrupulously to reveal the role of trehalose in rice under HS. In the revised MS, we have revised the title, abstract, and conclusion. By the way, the interpretation of the F1Fo-ATPase and ATP content was added in the discussion part which has been highlighted in purple. Thank you again.

Minorly,

2.6

Q2: What is the role of F0F1-ATPase?? Any relation between F0F1-ATPase and ATP content? Lack of interpretation…..

A: Thank you very much for your valuable suggestion. In the revised MS, the interpretation of the F1Fo-ATPase and ATP content was added in the discussion part which has been highlighted in purple. Thank you again.

Q3: What is 2nd para of discussion

SUS, SUT- are they short forms of genes? What is the full name? italic form?

A: Thank you very much for your valuable suggestion, and we are sorry for these careless errors. In the revised MS, the full name of these genes and italic form was added which has been highlighted in purple. Thank you again.

Q4: 3rd para

Overexpressing TPS >italic?

A: Thank you very much for your valuable suggestion, and we are sorry for these careless errors. In the revised MS, the italic form of TPS genes was added which has been highlighted in purple. Thank you again.

Reviewer 2 Report

I really appreciate authors positive efforts on the manuscript. But I still find some typo mistakes. Please carefully check grammar and spelling.

Good luck!

Author Response

Comments and Suggestions for Authors

I really appreciate authors positive efforts on the manuscript. But I still find some typo mistakes. Please carefully check grammar and spelling.Good luck!

A: Thank you very much for your positive evaluation, and we are sorry for the grammar and spelling errors. In the revised manuscript, we have checked the manuscript carefully, and also ask some other scholors for help. Hope to eradicate all the errors. Thank you again.

Round 3

Reviewer 1 Report

Well done, Congratulations!